# Influence of Short Carbon Fibers on the Properties of Autoclaved Fiber Cement in Standard Fire Environment

**DOI:** 10.3390/ma16062513

**Published:** 2023-03-22

**Authors:** Tomas Veliseicik, Ramune Zurauskiene, Modestas Kligys, Mark Dauksevic

**Affiliations:** 1Faculty of Civil Engineering, Vilnius Gediminas Technical University (VILNIUS TECH), 10223 Vilnius, Lithuania; 2Fire Research Centre, 13221 Valčiūnai, Lithuania

**Keywords:** carbon fiber, cellulose fiber, autoclaved fiber cement, standard fire curve

## Abstract

In case of a fire, the flame can spread from the building through the outer openings to the outside. In such cases, the fire temperature thermal effect determines the façade fibrocement tile thermal destruction, while the flammable thermo-insulating systems used for building energy effectiveness ensures it sets on fire. The spread of such a fire becomes uncontrollable and raises an immediate danger to the people inside the building, while such event dynamics delay and make it harder to put out the fire. Extra additive usage in façade fibrocement tiles can raise its resistance to fire temperature effect. Carbon fiber is widely known as a material resistant to the high temperature destructive effect. An investigation was conducted on the influence that carbon fiber has on the properties of autoclaved fiber cement samples. The autoclaved fiber cement samples were made from the raw materials, typical for façade fiber cement plates, produced in an industrial way (using the same proportions). In the samples, carbon fiber was used instead of mix cellulose fiber in 0.5%, 0.75%, 1% proportions. After completing the density research, it was determined that the carbon fiber effect had no general effect on the sample density. Ultrasound speed spreading research showed that the carbon fiber insignificantly makes sample structure denser; however, after the fire temperature effect, sample structure is less dense when using carbon fiber. The results of both these investigations could be within the margin of error. Insignificant sample structure density rise was confirmed with water absorption research, which during the 1% carbon fiber usage case was lower by 4.3%. It was found that up to 1% carbon fiber usage instead of mix cellulose fiber creates a dense structure of autoclaved fiber cement samples, and the carbon fiber in the microstructure influences the mechanical properties of the autoclaved fiber cement samples. After using carbon fiber in ambient temperature, the sample compressive strength and bending strength increased. However, the results of mechanical properties were completely different after experiencing fire temperature effect. Scanning electron microscopy research showed that the bond between the carbon fiber and the cement matrix was not resistant to high temperature effect, due to which the structure of the samples with carbon fiber weakened. Research showed that carbon fiber lowers the mechanical properties of the autoclaved fiber cement samples after high temperature effect. After analyzing the density, ultrasound speed spreading, water absorption, microstructure and macrostructure, compressive strength, and bending strength, the authors determined the main CF usage for AFK dependencies: 1. CF usage up to 1% replacing MCF makes the AFK structure more dense up to 1.5%, and lowers the water absorption up to 4.3%; 2. CF incorporates itself densely into the AFC microstructure; 3. CF usage up to replacing MCF improves the AFK strength properties up to until the fire temperature effect. Compression strength increases up 7.3% while bending strength increases up to 14.9%. 4. AFK hydrate amount on CF surface is lower than on MCF; 5. Fire temperature effect on AFK with CF causes dehydration by removing water vapor from the microstructure, resulting in a lot of microcracks due to stress; 6. The CF and cement matrix contact zone is not resistant to fire temperature effect. SEM experiments were used to determine the CF “self-removing” effect; 7. Due to complex changes happening in the AFK during fire temperature effect, CF usage does not improve strength properties in the microstructure. Compression strength decreases to 66.7% while bending strength decreases to 20% when compared with E samples.

## 1. Introduction

Previously carried out scientific research has shown how important it is to try and achieve the best mechanical properties of autoclaved fiber cement (AFC) in a fire effect temperature [1,2]. Cellulose fibers are usually used in AFC compositions. The main goal of their usage is structure reinforcement and even inner stress spread. During a fire, inner stress can form in the AFC due to high temperature effect, and to compensate for them, reinforcement material has to be resistant to the fire-created temperature effect. Scientists have noted a lot of positives of natural fiber, among which is their ecology, price, availability, etc., [3]. However, natural fibers also have negatives, such as their lack of resistance to high temperatures. MCF is chemically composed from cellulose, hemicelluloses, lignin, and sometimes other additives (wax, etc.). Hemicellulose thermal destruction starts near 200 °C, while cellulose component thermal destruction starts near 220 °C [4]. Since the cellulose component in natural fibers guarantees strength properties [5], the sample strength properties also worsen once they reach the mentioned temperature. Research carried out by scientists offers to use CF in cement materials, which would raise material strength properties when affected by higher temperatures [6]. This has an effect on AFC strength properties in fire effect temperature.

Carbon fiber (CF) is commonly used in production and can be found in various areas: aircraft production, vehicle production, sport and various household item production, and civil engineering [7,8]. The use of CF in building structures has led to scientific research on the behavior of such structures under various conditions [9,10,11]. CF as reinforcement additive compensates stress-created deformations when affected by high temperatures, it has a lower expansion coefficient and good thermal conductivity, is non-flammable, and can improve material strength properties. CF melting point temperature is above 3000 °C, which is significantly higher than the thermal destruction of cellulose fiber [12]. In addition to all the mentioned positive properties, CF has some disadvantages such as higher cost when compared to other fibers, and poor CF adhesion with cement matrix. In that case, if CF does not form a strong bond with the cement matrix, such reinforcement is ineffective. Seeking to increase the adhesion in contact zone between CF and cementing materials, several methods are used: mechanical, chemical (modifying the fiber), and using polymeric additive [13,14,15]. Liu [16] researched that CF in certain portions improves the mechanical properties of the cement matrix. Authors of [17] investigated the physical and mechanical properties of the cement material with CF and other additives and determined that CF usage in certain proportions can have a positive result. Authors of [18] investigated the CF effect on autoclaved materials and determined that increasing CF adhesion can give better physical and mechanical properties of autoclaved materials. Pehlinvanlı [19] showed that when studying autoclaved cement materials, CF improves the sample’s thermal conductivity, on which the thermal resistance of the material depends. Moreover, the investigation showed that CF increased the compressive strength of the samples.

Analyzing the literature showed that there were fewer research related to the determination of mechanical properties (in high temperatures) of cement materials when CF was used. Guo [20] investigated the influence of length and amount of CF on the mechanical properties of samples affected by high 400 °C and 800 °C temperatures. It is important to mention that scientists picked a temperature regime according to ISO 834-1 [21] standard which simulates the temperature regime created by fire. High temperature effect research was planned in three stages, holding samples for 2 h in determined 400 °C and 800 °C temperature. About 189 samples were investigated in the high temperature. Results showed that length and amount of CF do not have significant influence on the mechanical properties of samples when affected by a high temperature. Usually, CF was positively evaluated and most scientific works showed that it has a positive effect on the strength properties of researched materials [17,22,23]. However, there exists scientific research confirming conflicting results. Guo [20] determined that compressive strength in samples with 0.2%, 0.6%, and 1.4% of CF were lower when compared with the control samples (in both regular conditions and after high temperature effect). Such tendencies were also determined with bending strength tests. In this case, the samples with 0.2% and 0.6% of CF showed lower bending strength values compared with the control samples. In all mentioned experiments, the strength properties of cement materials and their dependency on the used amount of CF were determined. Scientific research analysis showed that the influence of CF on the physical and mechanical properties of cement materials was thoroughly investigated. However, research on the effects of CF on the AFC samples that have a complex composition (very close to production) and when they were affected by high temperature, do not exist. Currently, the influence of CF on the micro and macro structure formation and behavior when AFC samples were affected by a high temperature was not certain. The influence of CF on the mechanical properties of AFC samples is unknown. These abovementioned dependencies are attempted to be determined in this work, and the obtained research results are presented.

## 2. Materials and Methods

### 2.1. Materials

The length of short chopped CF, produced by a Japanese company, was 6 mm. Water soluble sizing of CF was declared by the manufacturer. Typical bulk density of CF was 350 g/l. In AFC, a reinforced fiber made up of two types of cellulose fibers was used. First type cellulose fibers were shorter (5.45%) than the other types of cellulose fibers (1.64%) and formed a so-called “hand fan”. This kind or cellulose is attributed to the natural fiber group and wood cellulose subgroup [24]. The chemical composition of CF and cellulose fiber is presented in Table 1.

Macrostructure image of CF and cellulose fibers was obtained with an optical microscope. An investigation was carried out to research the shape and surface of reinforced materials, which could have an impact on the formation of AFC matrix. Figure 1 shows CF and cellulose fibers shapes after optical microscope research.

Microstructure image of CF and a mix of cellulose fiber (MCF) was obtained using a field emission scanning electron microscope. An opportunity to research an uncovered electrically conductive layer surface and to perform an investigation on the changes appearing on the chosen surface place due to the heating arises. Scanning electron microscope research results are presented in Figure 2. When comparing CF to MFC visually, it can be noted that their geometry is different. First, MCFs are bigger than CF, later sample microstructure investigations show a clear visual difference between the fibers. It can also be seen that MCF fibers have an uneven geometric form, and they intersect with one another into various directions. CF fibers on the other hand are of a straight long form, visually looking rigid. Fiber intersecting can be seen in CF as well; however, it is easily traceable and less confusing when compared to MCF.

Continuing earlier experiments [2] as a base, the same raw materials and their combination were used, which makes it closer to composition, used in the facade fiber cement tile production (mixture marking E). CF was added to the mixtures partially, replacing thermally less-resistant MCF to improve the behavior of AFC samples in a high temperature and their properties after a high temperature effect. The amount of raw materials in the different mixtures presented in Table 2. 0.5%, 0.75%, and 1% of MCF was replaced (by mass) with CF in the mixtures, marked as AP050, AP075, and AP100.

Tap water (temperature 20 ± 2 °C) was used for the preparation of these mixtures. Water to cement ratio in the mixtures was 0.3. Portland cement CEM I 42.5 N, produced by a Lithuanian company and complying with EN 197-1 requirements, was used as the binding material. The mineralogical composition of clinker (Bogue calculation): C_3_S—68.1%; C_2_S—13.8%, C_3_A—8.6%; C_4_AF—7.1%. Quartz sand (M32) with specific surface of 287 cm^2^/g from JSC “Anyksčių kvarcas“ (Anyksciai, Lithuania) composed 99.5% silicate dioxide and 0.5% other admixtures (Al_2_O_3_, Fe_2_O_3_, TiO_2_, K_2_O, CaO). The same quartz sand was milled (M300) in a circular mill until it reached a specific surface 4400 cm_2_/g. Aluminum hydroxide and kaolin are additives, which during the autoclave fibro cement production process work in a complex way. In the scientific literature, several of their main usage aspects are mentioned: reducing moisture movement, reducing their carbonization rate, reducing stiffness and increasing their creep rate. Moreover, Al additive rises aluminum hydrosilicate amount in product [25]. Milled sand (M300) and cellulose fiber were prepared in a factory which produces autoclaved façade fiber cement plates. Cellulose fiber was factory separated in water (stored in closed buckets). The amount of water in the cellulose fiber was determined and the total amount of water in the mixture was adjusted before the mixing. CF was added at the start of the mixing by mixing it with milled sand (M300).

Mixtures were mixed in the laboratory mixer (Hobart type) with a forced blade movement, mixing mass at the speed of 45 rpm. First, CF and milled quartz sand (M300) were mixed for 3 min. Then the other raw materials were added in such order: quartz sand (M32), Portland cement, aluminum hydroxide, kaolin, MCF, and water. After adding all the row materials, the forming mixture was mixed for and additional 5 min. Trying to maintain this process similar to the industrial production of autoclaved facade fiber cement plates, the prepared mixtures were poured into the metal prism shaped forms (sized 40 mm × 40 mm × 160 mm) and compacted on a vibrating table, which is used for mortars. About 0.5 kg load was used on the top of the samples during the compaction. Samples were kept in the metal forms for 8 h and afterwards the samples with the forms were put into the autoclave. Samples were hardened in the autoclave for 10.5 h with pressure 10.13 bar and temperature 179 °C. The rising of the pressure in the autoclave was performed for 2 h, the samples were held in the highest pressure for 7 h, and the pressure was released for 1.5 h. The samples were held in the environment of 60% humidity for 14 days, after hardening in the autoclave.

### 2.2. Methods

The investigations of the macrostructure of samples were carried out by an optical microscope “Leica M165 FC“. The investigation of microstructure of CF was carried out with the field emission scanning electron microscope EVO LS 25, Zeiss Germany (accelerating voltage–20.0 kV).

Samples for the dry density determination were dried in a laboratory climatic chamber at 100 °C ± 5 °C temperature until constant mass was reached. Samples were conditioned according to the requirements of standard EN 13238 [26] in a conditioning room at 22 °C ± 2 °C temperature and 50% ± 5% humidity, when determining the sample air dry density. Sample absorption kinetics were determined by soaking them in water of +20 °C ± 2 °C temperature. Samples were dried before the experiment in a laboratory climatic chamber at 100 °C ± 5 °C temperature until a constant mass was reached. Water absorption starting periods were determined to be 0.5 h, 1 h, 3 h, 6 h, 12 h, 24 h and after weighting the samples it was repeated for each day until the mass change of the sample was smaller than 0.1%. Before every weighting, the leftover water was wiped away from the samples. Experiment was carried out and the results were calculated according to the LST EN 826 standard (A method). Development of the structure in the samples was evaluated by means of ultrasonic pulse velocity method, using the tester “Pundit 7”. The values of ultrasonic pulse velocity were obtained after hardening and drying of the samples. Samples were placed between two ultrasonic transducers (transmitter and receiver) operating at a frequency of 54 kHz. The transducers were pressed against the samples at two strictly opposite points. Vaseline was used to ensure a good contact. The ultrasonic pulse velocity was calculated using this equation [27]:(1)$$V=\frac{l}{t}\times 10^{6}$$
where:

*l*—distance between cylindrical heads, m;

*t*—time of pulse spread, s.

The compressive strength test was conducted in the powerful electromechanical testing machine H200kU of capacity 200 kN and load measurement accuracy ±0.5% of applied load, from 0.2% to 100% capacity. The speed of load increment was 60.0 N/s until destruction of the sample. The experiment was carried out and results calculated according to the LST EN 826 standard. Concluding test result was taken as the average value calculated out of at least five successful measurements.

Bending strength testing equipment was used. At half-length of each sample the linear load was applied across the width (3 bending points). The speed of increment of bending load was 14 N/s. The final test result was an average value calculated out of at least three successful measurements. The experiment was carried out and results were calculated according to the LST EN 12089 standard.

Sample temperature was affected using a heating chamber according to the standard fire curve (ISO 834), which describes the Formula (2) [21]; graphic expression is shown in Figure 3. This kind of simulation of fire dynamic is used in the standardized construction fire test [28,29]. Moreover, this kind of fire environment simulation is used in scientific research [30]. Samples were added into the heating environment in groups of three, while making sure each side of the sample surface would be affected by the temperature. After 30 min, the sample temperature reached 842 °C. The experiment was stopped and the samples were cooled down in the natural way. After cooling them to the laboratory room temperature, samples were carried over to the conditioning room with 22 °C ± 2 °C temperature and 50% ± 5% humidity.
(2)$$\theta_{g}=20+345\;\log_{10}\left(8t+1\right),$$
where

*θ_g_*—fire temperature, °C; 

*t*—time, min.

**Figure 3 materials-16-02513-f003:**
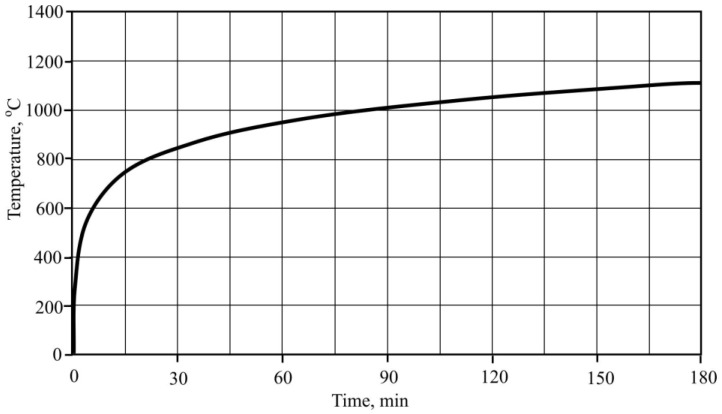
Standard fire curve.

## 3. Results

It is known that CF is essentially an inert additive that does not affect the formation of minerals and compounds during the hardening process of cementitious materials. Like most fibrous materials, CF is usually used in cementitious materials for their reinforcement [31]. Therefore, during this study, the mineral composition of the samples and its change depending on the used amount of CF were not investigated.

During the investigation it was determined that CF does not essentially have the influence on the dry density of the samples. It can be seen that the maximum difference between the samples was only 4 kg/m^3^, which could be attributed to the margin of error. After carrying out the dry density determination in the samples affected by the fire temperature it was fixed that the dry density of all the samples decreased. Depending on the amount of CF, which in samples was from 0.5 to 1%, the dry density of the samples changed from 1360 kg/m^3^ to 1375 kg/m^3^. For comparison, the dry density of sample E was 1380 kg/m^3^. The results of dry density are shown in Figure 4.

The results of ultrasonic pulse velocity measurements are shown in Figure 5. It can be seen that increasing the amount of CF in the samples insignificantly increases the ultrasonic pulse velocity. The highest increase was determined in the AP100 samples. Ultrasonic pulse velocity was determined to be 3364 m/s when compared to sample E (3288 m/s), which comprises 2.3%. Increase in the percentage of the ultrasonic pulse velocity in the other samples was even lower, compared with the E samples. Such an insignificant effect on sample E can mean that CF does not have a high influence on the ultrasonicpulse velocity. The tendency of the results of ultrasonic pulse velocity in the samples after fire effect was different compared to the samples before fire effect. When compared with sample results which were obtained from the fire temperature effect, in this case the determined difference was lower. The determined ultrasonic pulse velocity in AP100 sample was 3063 m/s and that was 1.7% lower than in sample E. Determined ultrasonic pulse velocity in samples AP050 and AP075 was lower than that of sample E.

Sample absorption kinetics were determined and shown in Figure 6. Research results showed that the samples with CF had lower water absorption when compared with sample E. The difference between the investigated samples was not high, it reached 4.3%. These results can be explained with natural cellulose fiber water absorption properties, which in most cases was better than those of synthetic fibers [32], while the low amount of changed additives shows insignificant changes. A small influence on that can also be due to the densifying effect of CF in the structure of the samples.

Research of samples surface which was carried out by optical microscope techniques showed macro-structure formation without adding CF (Figure 7a) and showed CF traces before (Figure 7b) and after fire effect (Figure 7c). Research showed that CF after fire effect remains in the matrix, when compared to MCF. 

Research of microstructure of samples before fire temperature impact showed that CF can be compactly inserted into AFC composite structure (Figure 8). With the current MCF they intertwine in all directions, thus strengthening AFC composite. Moreover, scanning microscopy studies revealed that the formation of hydrates on MCF is higher than on CF (Figure 9). 

Research of microstructure of samples after fire temperature impact showed that in samples with CF, the amount of microcracks was higher than that in E samples (Figure 10 and Figure 11).

Research results showed that compressive strength of the samples AP050, AP075, and AP100 was higher than that of sample E (Figure 12). The highest compressive strength was determined in sample AP100, and was 7.3% higher than in sample E. Similar results were obtained in both the samples, AP050 and AP075. The determined compressive strength in these samples was accordingly 1.58% and 5.92% higher than in sample E. Therefore, changing the amount of CF in the samples from 0.5% to 1% compressive strength increased from 51.5 to 54.4 MPa when compared with sample E (50.7 MPa).

After 30 min standard fire effect, compressive strength determination was carried out using the experiment samples. Compressive strength of the samples AP050, AP075, AP100 decreased in a similar range and were approximately from 49.5% to 66.7% lower compared to the results of the sample E. As the amount of CF in the samples varied from 0.5 to 1.0%, compressive strength of the samples after fire effect varied from 18.5 MPa to 12.2 MPa.

Results of bending strength of the samples (Figure 13), as well as compressive strength results showed that bending strength of the samples AP050, AP075, AP100 was higher than sample E. The highest bending strength was determined in the samples AP050 as well and it was 14.9% higher than in sample E. Bending strength of the samples AP075 and AP100 was 8.5% and 6.4% higher respectively. Bending strength of the samples with CF (in natural conditions) was from 5.4 MPa to 5.0 MPa.

After 30 min of standard fire effect, bending strength determination was carried out using the experiment samples. Bending strength of the samples decreased from 46.8% to 60.0% compared to the results of sample E. Decrement of bending strength of samples AP050, AP075, AP100 was the same (about 60.0%). After the fire effect bending, strength of the samples with CF decreased from 2.2 MPa to 2 MPa. It was lower than 2.5 MPa in sample E.

## 4. Discussion

Systematic interpretation of the obtained results starts from micro and macro structure formation. Since it was known that CF does not form new formations in cement materials during hydration, for appropriate structure formation CF mixing solutions were used. The fiber and cement matrix contact zone can fuse in two ways: by modifying the cement matrix or by modifying the fiber [33]. Quartz sand amount is usually higher than CF [34], due to this sample mixing starts with quartz sand (M300) mixed with CF. It was aimed that on CF surface, incisions would form that would increase CF and cement matrix adhesion [35]. However, when carrying out scanning electron microscope research, no abrasions or cement matrix formation centers around CF were fixed (Figure 8 and Figure 9). Researchers in this work evaluated only the cement matrix modification and formed a contact zone between CF and MCF. Cement matrix was modified during hydrothermal hardening due to the use of aluminum hydroxide. Tobermorite is the main CSH group hydrate, formed in autoclave fibrocement material [1,36]. Sample microstructure research up to fire effect showed that the hydrate amount which forms on different types of fibers is different. Even though CF can have a higher surface area, flatter surface structure determined that practically almost no hydrates formed on it when compared with MCF (Figure 8 and Figure 9). From that it can be determined that the CF and cement matrix contact zone is weaker than the MCF and cement matrix contact zone.

Sample macro-structure research after the fire effect showed that MCF burns out leaving channels in the cement matrix, while CF remains unchanged. Visually noted thermal cracks in the sample surface are shown in Figure 7c. Research shows that after the fire effect CF remains in sample cement matrices. Sample with CF microstructure research after fire showed that in cement matrix, besides leftover CF, burned out MCF channels and loads of microcracks appeared (Figure 10). In the samples, CF crack amount was higher when compared with samples which used only MCF (Figure 11). Seeking to determine the reason behind this mechanism, it would be worthwhile to evaluate other research which are associated with structure formation.

Since the amount of changed fibers in the compositions of AFC samples was insignificant and did not compose more than 1% from the main mass, obvious differences in density of the samples were not determined. It is known that cellulose density is about 1.5 g/cm^3^ [37], while CF density can be 1.74–2.19 g/cm^3^ [38]. Higher volume and lower density MCF were changed to lower volume and denser CF. Usually, the sample structure would densify using this method; however, the changed amounts were low. Even though density results showed an insignificant tendency for density to be lower, it cannot be observed as an obvious dependency due to the margin of error. Same results of density were determined in Zegardło [39] research. Sample density research results from fire effect did not present obvious tendencies; however, this could be explained by changed fiber amounts being low. Sample density experiment results after fire temperature showed regular density decrease results for all samples. Density decrease is associated with hydrate dehydration processes and MCF burning out. When comparing sample densities together, even though there is a determined tendency of sample density after fire lowering when increasing CF amount, this difference is low and changes within margins of error.

More clear sample structure change results were presented by ultrasonic pulse velocity measurements. Increased amount of CF in the samples before a high temperature effect influenced higher ultrasonic pulse velocity values. This may be related to CF size which is more compact than MFC. In such a case, CF densifies structure of the samples more than MCF, while also the previously mentioned density of fibers has an influence on the results. It cannot be claimed that increasing the amount of CF obviously makes AFC sample structure denser, since the results can be evaluated to be within the margin of errors; however, quantitative sample result values stand out when compared with the results of density measurements. E samples with only MCF have a higher water absorption when compared with CF. Water absorption research confirms that CF makes sample structure denser.

After high temperature effect, it was determined that the samples with CF have a lower ultrasonic pulse velocity than the sample E. It is known that CF is resistant to fire temperature effect when compared with MFC. In high temperature effect, MFC thermal destruction happens which determines destruction of material structure and possibly the appearing thermal fractures; however, ultrasonic pulse velocity results showed that the microstructure of the samples with a higher amount of MFC was denser. This correlates with the microcracks in samples, due to which ultrasonic pulse velocity lowers.

Analyzing the microstructure revealed that microcracks in samples with CF appeared to be denser than in sample E microstructure. During the fire temperature effect, besides starting hydrate splitting in samples, water is also dehydrated. In sample with CF, higher density creates an obstacle for water vapor removal. When stress that breaks down the sample cement matrix increases, stress can also appear due to different fiber usage when their coefficient of thermal expansion is different. Samples only with MCF, due to its thermal destruction and more homogenic structure, had a lower amount of microcracks.

It was determined that in samples up to the fire effect, the hydrate amount on MCF is higher than that of CF samples. It is possible that such an influence might be caused not only by the appropriate modification of cellulose fiber in the cement matrix, but also MCF water absorption. Water absorbed by MCF, in terms of time, spreads more evenly during the cement hydration process. It is known that pre amount has an influence toward cement material mechanical properties. Microstructure research showed that AP050, AP075, AP100 samples after strength property experiment part CF “breaks”. This “break” shows that up to fire temperature effect in some places CF forms a stronger contact zone with the cement matrix (Figure 14).

Contact zone between fibers and cement matrix after fire was researched using SEM method. The present research showed that CF does not form a strong bond with cement matrix and depending on the effect, CF is “self-removing” from the cement matrix (Figure 15). The CF “self-removing” effect is known and in 1990 scientists described this effect mechanism in cement materials [40]. Microstructure research results showed that CF densities incorporate into the microstructure (Figure 8); however, on their surface the formed hydrate amount is not that high when compared with MCF. During the AFK fire effect, complex changes take place such as removing water and creating extra stress; the CF surface starts to oxidize at a temperature higher than 600 °C [41]; and dehydration of the main hydrate (CSH) takes place [2]. Such complex effect determines that the contact between CF and cement matrix decreases.

Results of compressive strength of the samples in environment temperature showed that CF slightly increased the compressive strength. The most obvious compressive strength increase was determined in the samples AP100 and it was higher by 7.3% than in the sample E. The formation of dense microstructure has a high influence on the compressive strength. During AFC composite hardening, CSH (tobermorite), katoite, hydrogarnet, and other hydration products form, and MCF and CF fit into the microstructure; and when a strong bond between MCF and cement matrix appears, an additional bond between CF and cement matrix appears, and the needed effect is reached. These essentially are some of the main reasons why the compressive strength results slightly increased. Results of bending strength had similar tendencies and regularity when compared with the results of compressive strength. Environment temperature samples with CF had a higher bending strength than the sample E. It was mentioned before that CF fits into dense composite structure where it also intertwines with MCF (Figure 8). Such complex usage of various fibers gives better bending strength results, even though CF mixing with quartz sand did not affect the CF surface. However, CF fiber is not as flexible as MCF, due to which higher usage of CF in samples AP075 and AP100 does not give better bending strength results compared to that in AP050 samples. Bending strength of the samples AP100 was 14.9% higher when compared with the sample E.

Results of compressive strength of the samples after fire temperature effect showed that compressive strength decreased when increasing the amount of CF. In general, this was due to dehydration, micro and macro fractures, opened pores, and other phenomena specific to cement materials which appear when affected by a high temperature. However, when comparing results of the sample E with the samples with CF it can be stated that the latter sample values decreased. The highest decrease (54.3%) was in sample AP100, which had the highest amount of CF. Sample results with higher CF amount were determined by higher microcracks amount, which during the fire temperature effect was conditioned by the release of water vapor from the sample. Such a destructive effect of course had a significant effect on sample strength properties. Furthermore, not a particularly strong and not thermally resistant contact zone between CF and cement matrix further weakened the microstructure. Results of bending strength of the samples after fire temperature effect decreased and their significant decrease was in the samples with CF. The highest decrease (60%) was in the sample AP100, which had the highest amount of CF. Essentially, bending strength results and their tendencies are similar with compression strength sample results, due to the cement matrix destruction mechanism remaining the same. A tendency can be seen that the higher the amount of CF, the higher the decrease in compressive strength.

## 5. Conclusions

It was determined that in regular conditions CF does not essentially influence the dry density of the samples. After fire temperature exposure, research results showed a slight decrease in dry destiny in samples with CF. It was established as 1.5% when compared with the E samples.

Ultrasonic pulse velocity tests results showed insignificant increases when adding CF. The highest increase was 2.3% and ultrasonic pulse velocity was rising with adding CF. Opposite results were obtained with samples after exposure to fire. In this case, highest ultrasonic pulse velocity was in samples where the amount of CF was lower and it was decreasing with adding CF. Ultrasonic pulse velocity in samples with 1% of CF was 1.7% lower than in E samples.

Sample absorption kinetics research was carried out in regular conditions and research results showed that the samples with CF had lower water absorption when compared with E samples. The difference between the investigated samples was not high, it reached 4.3%.

It was determined that CF, partly replacing MCF, in regular conditions improved the mechanical properties of the AFC samples. The CF evenly spread and fitted into AFC structure, due to which compressive strength slightly increased. The highest increase was of 7.3% when compared with results of the samples without CF. After 30 min fire effect, all samples with CF had lower compressive strength than samples without CF, which only had MCF. The highest decrease was an astounding 66.7% when compared with the samples where the reinforcement material was only MCF. The CF made AFC structure denser, but in case of high temperature, did not compensate for microstructure shrinkage.

Results of bending strength showed similar tendencies to the ones of compressive strength. Samples with CF in environmental conditions showed better bending strength results when compared to the samples without CF. Reinforcing CF fitted with MCF into AFC structure increased the bending strength by 14.9% when compared with the control samples. Bending strength experiments carried out after high temperature effects showed that CF significantly decreased the bending strength. The highest bending strength decrease was 20% when compared with the samples where the reinforcement material was only MCF.

Scanning electron microscope research showed that CF evenly spreads in the mixture and forms a dense structure; however, it does not create a strong bond with the cement matrix. In high temperatures, the adhesion in the contact zone between CF and cement matrix decreases. During the research, CF “self-removing” from AFC structure were determined.

Authors determined the main CF usage for AFK dependencies:CF usage up to 1% replacing MCF makes the AFK structure more dense up to 1.5% and lowers water absorption up to 4.3%;CF incorporates itself densely into the AFC microstructure;CF usage up to replacing MCF improves the AFK strength properties up to until the fire temperature effect. Compression strength increases up 7.3% while bending strength increases up to 14.9%.AFK hydrate amount on CF surface is lower than on MCF;Fire temperature effect in AFK with CF causes dehydration by removing water vapor from the microstructure, resulting in a lot of microcracks due to stress;The CF and cement matrix contact zone is not resistant to fire temperature effect. SEM experiments were used to determine CF “self-removing” effect;Due to complex changes happening in AFK during fire temperature effect, CF usage does not improve the strength properties in the microstructure. Compression strength decreases to 66.7% while bending strength decreases to 20% when compared with E samples.

## Figures and Tables

**Figure 1 materials-16-02513-f001:**
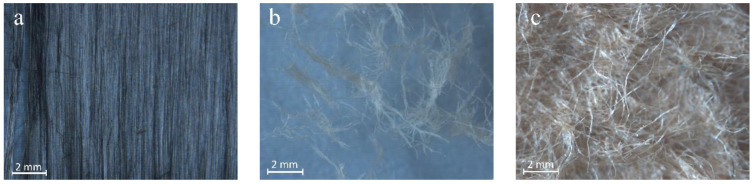
Images (zoomed in 25 times) of CF (**a**), shorter cellulose fiber (**b**) and “hand fan” cellulose fiber (**c**).

**Figure 2 materials-16-02513-f002:**
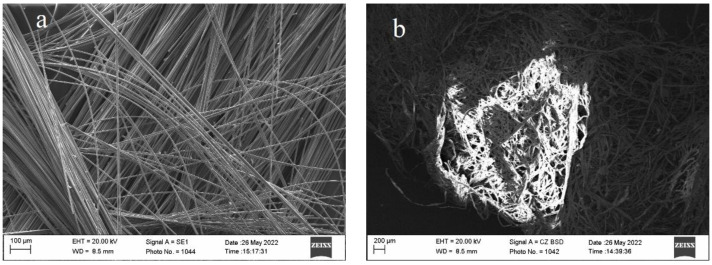
Scanning electron microscope image of CF (**a**) and MCF (**b**).

**Figure 4 materials-16-02513-f004:**
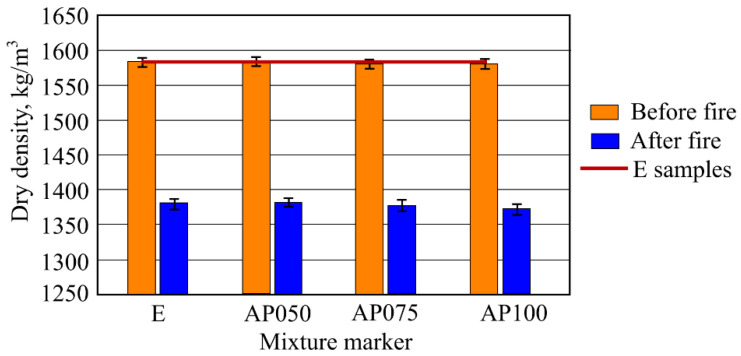
Results of dry density (before and after fire effect) of the samples.

**Figure 5 materials-16-02513-f005:**
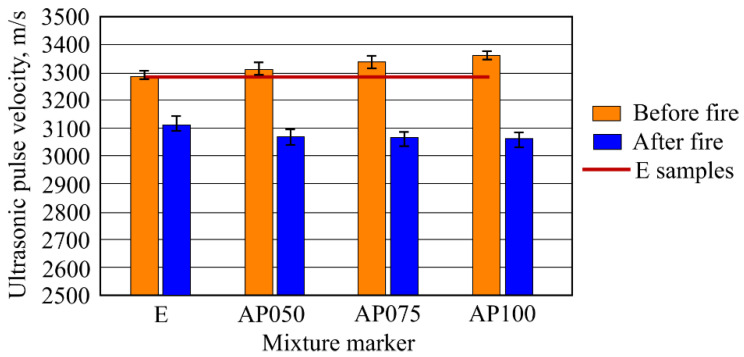
Results of ultrasonic pulse velocity (before and after fire effect) of the samples.

**Figure 6 materials-16-02513-f006:**
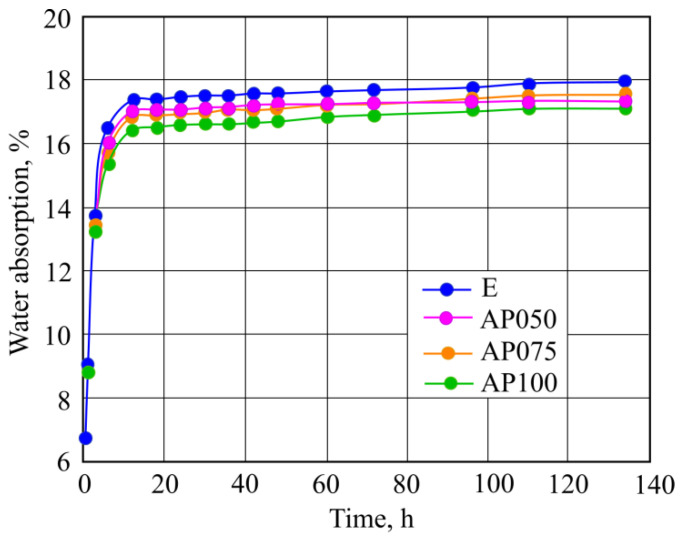
Results of water absorption of the samples.

**Figure 7 materials-16-02513-f007:**
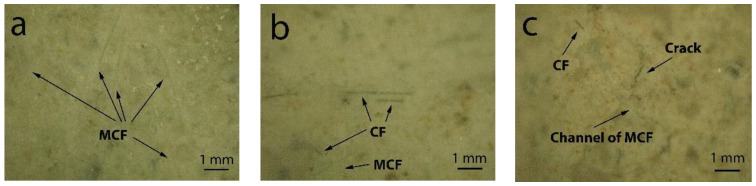
Macrostructure of the samples zoomed in 100 times: (**a**) without adding CF (E sample); (**b**) after adding CF (AP050 sample); (**c**) after fire temperature exposure (AP050 sample).

**Figure 8 materials-16-02513-f008:**
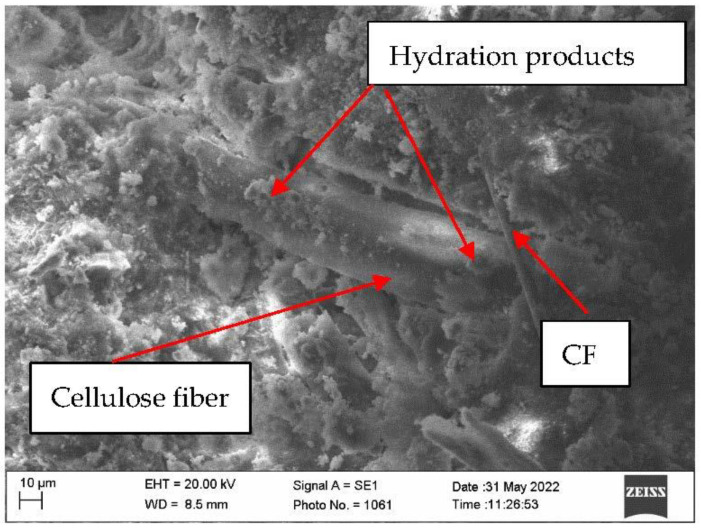
Microstructure of the sample AP050.

**Figure 9 materials-16-02513-f009:**
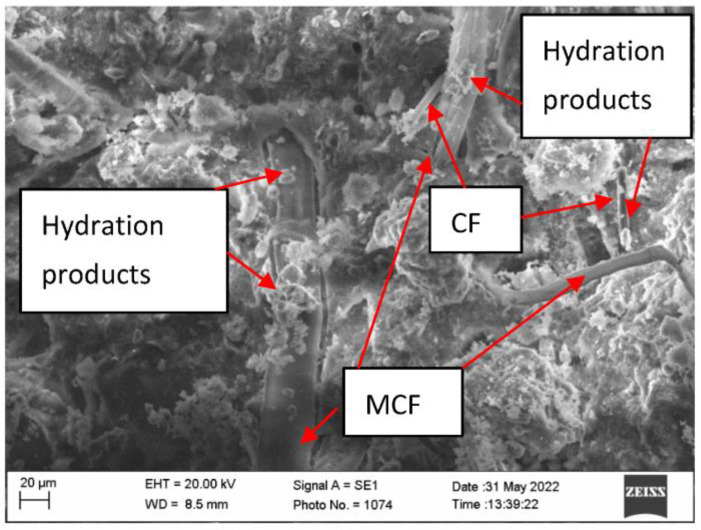
Microstructure of the sample AP100 before fire exposure.

**Figure 10 materials-16-02513-f010:**
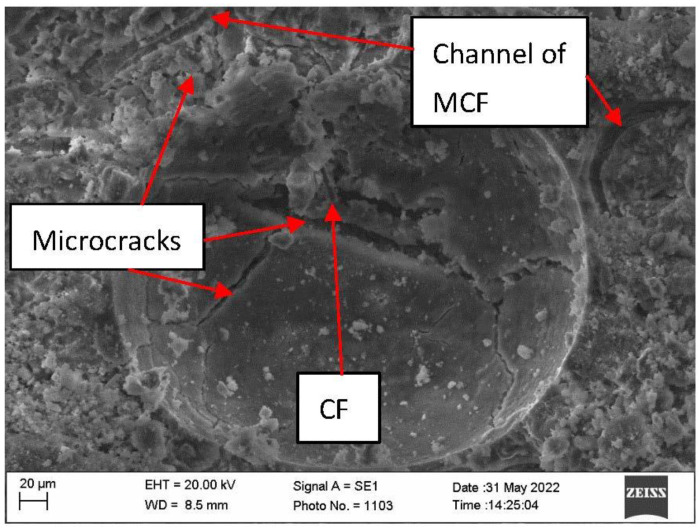
Microstructure of the sample AP100 after fire exposure.

**Figure 11 materials-16-02513-f011:**
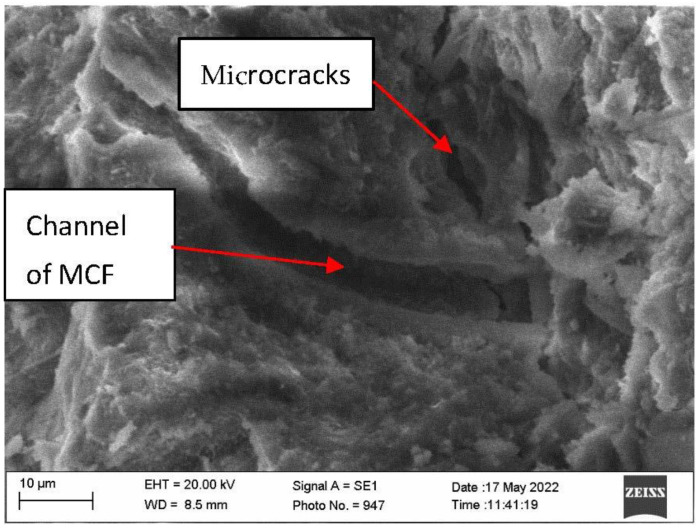
Microstructure of sample E after fire exposure.

**Figure 12 materials-16-02513-f012:**
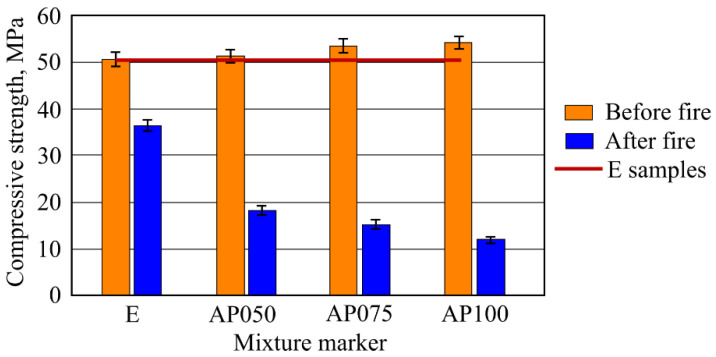
Results of compressive strength (before and after fire effect) of the samples.

**Figure 13 materials-16-02513-f013:**
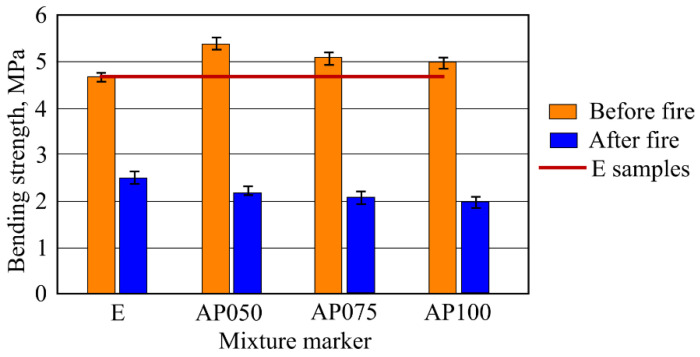
Results of bending strength (before and after fire effect) of the samples.

**Figure 14 materials-16-02513-f014:**
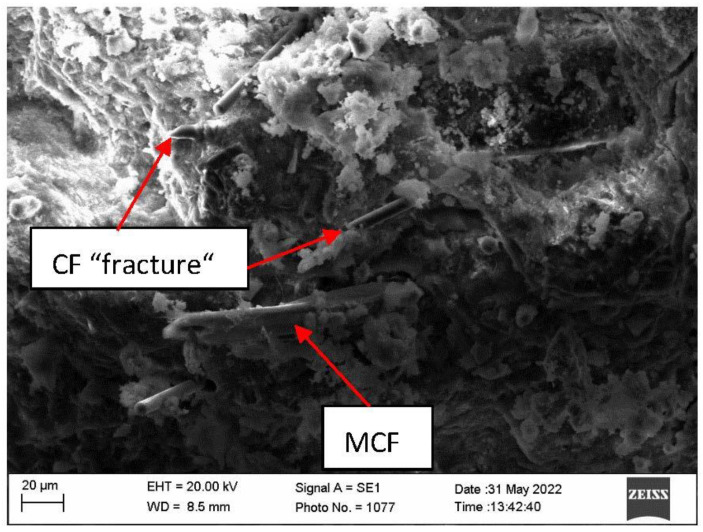
The CF “fracture” from cement matrix due to mechanical effect in AP100 samples before fire temperature effect.

**Figure 15 materials-16-02513-f015:**
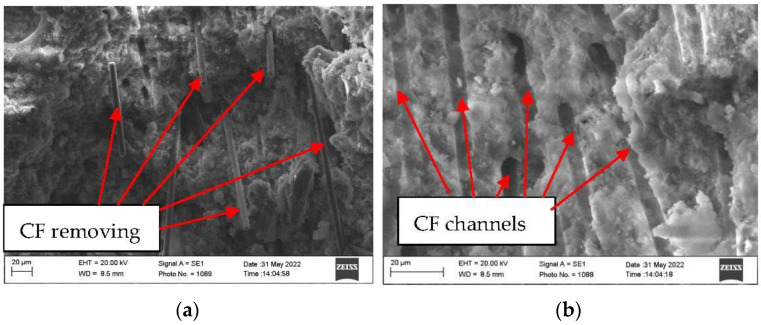
The CF “removal” from cement matrix due to mechanical effect in samples after fire temperature effect: (**a**) CF residue, (**b**) remaining channels after CF removal.

**Table 1 materials-16-02513-t001:** Chemical composition of CF and cellulose fiber.

Material	Chemical Element Content, Weight %						
	O	C	S	Si	K	Ca	Total
CF	0.1	99.9	–	–	–	–	100
Cellulose fiber	48.3	50	0.4	0.1	1	0.2	100

**Table 2 materials-16-02513-t002:** The amount of raw materials in the different mixtures in %.

Mixture Markings	Sand M32	Sand M300	Portland Cement	MCF	Aluminum Hydroxide	Kaolin	CF
E	41.05	10	38.04	7.09	2.06	1.76	-
AP050	41.05	10	38.04	6.59	2.06	1.76	0.50
AP075	41.05	10	38.04	6.34	2.06	1.76	0.75
AP100	41.05	10	38.04	6.09	2.06	1.76	1.00

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
