# Peer review of "Influence of Short Carbon Fibers on the Properties of Autoclaved Fiber Cement in Standard Fire Environment"

_materials, 2023, doi:10.3390/ma16062513_

Round 1

Reviewer 1 Report

In this paper, an investigation was conducted on the influence that carbon fiber (CF) has on the prop- 10erties of autoclaved fiber cement (AFC) samples. Ultrasonic pulse velocity (UPV), dry density, porosity, optical microscope (OM) and scanning electron microscope (SEM) research methods were used to identify micro and macro structures of samples. Mechanical properties were evaluated using bending and compressive strength research methods. The research shows that the compressive strength and flexural strength of the sample are improved when CF replaces MCF in a certain proportion at room temperature. However, the mechanical properties of AFC samples are reduced by CF at high temperature. This work will definitely add valuable knowledge to the current research community, and hence deserves to be published. Nevertheless, the authors should address the following issues before its acceptance for publication.

1.In Section 3, there are two figures 10, so that the following graphic annotations are confused. Please correct it according to the specific content of the article.

2.In the second section of the article, "Materials and Methods", it will be more organized to describe it in two headings.

3.Line 133it is mentioned that the use of tap water (temperature 20 ± 2 ) to prepare these mixtures has any impact on the experimental results? If there is any impact, the situation should be explained in the text.

4. Some recent references should be added in the Introduction part.

5. The recent relevant research should be introduced in the manuscript. e.g., Durability life prediction and horizontal bearing characteristics of CFRP composite piles in marine environments. Construction and Building Materials, 2023, 367: 130116.

6. It is not clear whether the formulae involved in the article were derived by the authors themselves or from other literature? All equations that have no reference are from the authors?

7. In the text, experiments were carried out on the dry density, ultrasonic pulse velocity and water absorption of the samples with the addition of CF, but why is there no summary in the conclusion section? Please add it in the conclusion section.

Author Response

Dear Reviewer,

Thank You for comments and suggestions. We are sending for You revised manuscript and our answer for Your comments.  

1. Agree. Numeration of figures changed.

2. Agree. Second section of the article “Materials and Methods” organized in two heading.

3. Clarified. Usually in the research we use water from tap (like in technological process). We didn’t research influence water temperature on our samples characteristic.

4. Agree. We added new references to the introduction part.

5. Agree. Added in the manuscript.

6. Agree. Added references to all equations.

7. Agree. Added information about CF influence on the samples dry density, ultrasonic pulse velocity and water absorption in conclusions section. 

Reviewer 2 Report

Abstract and section Introduction is clearly written. Too much abbreviations is given in the abstract. Some of the least important could be given in the main text.

The background, current state of the art is described. A small paragraph dedicated to the fire performance of autoclaved cement could be given, as previous researches confirm that fiber does not give improved results which could be associated with the deterioration of the cement matrix.

2. materials and methods.

In fig.1, the scale factor is hardly visible. Please provide larger ruler.

A small description and discussion for images 2a and 2b could be introduced. Does measurements conform with manufacturers data, compare both fibers (geometry, appearance etc.).

Table 2. A small “reminder” could be introduced what is meant by M32 and M300 sands.

Please mention what is the role of aluminium hydroxide in the composition.

There are a lot of abbreviations used in the whole text. Maybe if there are less than 3-4 mentions in the whole text, it is worth to use full words. There are some abbreviations which are not used in the text (UPV - ultrasonic pulse velocity (check page 4, page 6).

Results.

What is meant as etalon in figure 4? It looks the same as sample weight before fire. Is there any deviation of the results available? Similar for figure 5.

In figure 7 and 8, please specify in caption and in text, what sample condition is it (before or after heat treatment).

Discussion is clearly written, while the conclusions somehow continue discussions and general finding from the research should be outlined there.

Author Response

Dear Reviewer,

Thank You for comments and suggestions. We are sending for You revised manuscript and our answer for Your comments.  

Abstract and section Introduction is clearly written. Too much abbreviations is given in the abstract. Some of the least important could be given in the main text.

Agree. Some abbreviations deleted from abstract, methods abbreviations deleted from manuscript.

The background, current state of the art is described. A small paragraph dedicated to the fire performance of autoclaved cement could be given, as previous researches confirm that fiber does not give improved results which could be associated with the deterioration of the cement matrix.

Agree. Added some new reference about autoclave fiber cement.  

  1. materials and methods.

In fig.1, the scale factor is hardly visible. Please provide larger ruler.

Agree. Provides larger ruler.

A small description and discussion for images 2a and 2b could be introduced. Does measurements conform with manufacturers data, compare both fibers (geometry, appearance etc.).

Agree. Added  small discussion for 2a and 2b images. 

Table 2. A small “reminder” could be introduced what is meant by M32 and M300 sands.

Agree. Added information about sands. 

Please mention what is the role of aluminium hydroxide in the composition.

Agree. Added short information about aluminium hydroxide in the composition. 

There are a lot of abbreviations used in the whole text. Maybe if there are less than 3-4 mentions in the whole text, it is worth to use full words. There are some abbreviations which are not used in the text (UPV - ultrasonic pulse velocity (check page 4, page 6).

Agree. Abbreviations deleted. 

Results.

What is meant as etalon in figure 4? It looks the same as sample weight before fire. Is there any deviation of the results available? Similar for figure 5.

Agree. Changed from etalon to E samples. E samples was without adding CF.

In figure 7 and 8, please specify in caption and in text, what sample condition is it (before or after heat treatment).

Agree. Added information about conditions. 

Discussion is clearly written, while the conclusions somehow continue discussions and general finding from the research should be outlined there.

Agree. Added general findings from the research to conclusions.

Reviewer 3 Report

The article is interesting for publication, but some points need to be answered:

- The abstract is very simple and does not present the research motivations. Also, the main results and main conclusions of the research are not presented. Please review this.

- The introduction needs to highlight the originality of the research. The authors argue: “Analyzing literature it became clear that there was not a lot of research related to the determination of mechanical properties (in high temperatures) of cement materials when CF were used. Guo [14] investigated the influence of length and amount of CF on the mechanical properties of samples affected by high 400 °C and 800 °C temperatures.” It is very unlikely that only this work is interesting to be cited in the research. Authors must present at least 4 or 5 previous contributions to highlight the originality of the research.

- The explanation about Cellulose fiber is very vague and totally flawed. Authors should delve deeper into this point. This is a recent research topic that has been gaining a lot of attention. Therefore, please explain what type of cellulosic fiber is used and go into more detail about this type of material. I suggest analyzing the following publications: 10.1007/s41062-021-00551-8 and 10.3390/polym14030647. Based on these articles, deepen your discussion on cellulosic fibers. Explain about heterogeneity and durability issues.

- What is Sand M32 and Sand M300? Please explain this nomenclature better.

- How were the compositions in Table 2 defined. Why were these amounts of cement, sand, water and fiber defined? Please explain properly.

- Explain a little more about Figure 3. If possible, include more recent references about the information present in this figure.

- What is Etalon present in Figure 4 and 5? How was this calculated? What represents? I don't know this information. Please delve deeper.

- Compare the results of Figures 4 and 5 with other similar studies already published. Please further your discussion of the information presented in this section.

- The discussions of Figures 7 and 8 are very bad. If you don't have discussions to do about these figures, it's more interesting to remove them from the article. Please discuss the information in this figure.

- The article is very simple. Discussions are very weak. Look at Figures 9 and 10. Discuss the information illustrated by the figures. Compare with similar searches. Include the deviation bar. Indicate the % of resistance loss after fire. Explain the mechanisms that take place in the fiber that cause these patterns of behavior. Please enhance your discussions.

- Review conclusions based on previous comments. If necessary, change the conclusions.

Author Response

Dear Reviewer,
Thank You for comments and suggestions. We are sending for You revised manuscript and our answer for Your comments.  

- The abstract is very simple and does not present the research motivations. Also, the main results and main conclusions of the research are not presented. Please review this.

Agree. Abstract was reviewed. 

- The introduction needs to highlight the originality of the research. The authors argue: “Analyzing literature it became clear that there was not a lot of research related to the determination of mechanical properties (in high temperatures) of cement materials when CF were used. Guo [14] investigated the influence of length and amount of CF on the mechanical properties of samples affected by high 400 °C and 800 °C temperatures.” It is very unlikely that only this work is interesting to be cited in the research. Authors must present at least 4 or 5 previous contributions to highlight the originality of the research.

Agree. Added some new reference. Also in manuscript were mentioned about lack of this kind research (autoclave fiber cement, fire and carbon fiber). 

- The explanation about Cellulose fiber is very vague and totally flawed. Authors should delve deeper into this point. This is a recent research topic that has been gaining a lot of attention. Therefore, please explain what type of cellulosic fiber is used and go into more detail about this type of material. I suggest analyzing the following publications: 10.1007/s41062-021-00551-8 and 10.3390/polym14030647. Based on these articles, deepen your discussion on cellulosic fibers. Explain about heterogeneity and durability issues.

Agree. Added publications to manuscript. Added more information about fiber. 

- What is Sand M32 and Sand M300? Please explain this nomenclature better.

Agree. Added information about sands. 

- How were the compositions in Table 2 defined. Why were these amounts of cement, sand, water and fiber defined? Please explain properly.

Revised. Information about amounts of materials in samples is based on industrial production and it was mentioned in manuscript. 

- Explain a little more about Figure 3. If possible, include more recent references about the information present in this figure.

Agree. Added more information about Figure 3. 

- What is Etalon present in Figure 4 and 5? How was this calculated? What represents? I don't know this information. Please delve deeper.

Agree. Changed from etalon to E samples. E samples was without adding CF.

- Compare the results of Figures 4 and 5 with other similar studies already published. Please further your discussion of the information presented in this section.

Agree. Added results from similar studies.

- The discussions of Figures 7 and 8 are very bad. If you don't have discussions to do about these figures, it's more interesting to remove them from the article. Please discuss the information in this figure.

Agree. Added deeper discusion of microstrukcture.

- The article is very simple. Discussions are very weak. Look at Figures 9 and 10. Discuss the information illustrated by the figures. Compare with similar searches. Include the deviation bar. Indicate the % of resistance loss after fire. Explain the mechanisms that take place in the fiber that cause these patterns of behavior. Please enhance your discussions.

Agree. Added deeper discusion of microstrukcture, mechanisms was explained. 

- Review conclusions based on previous comments. If necessary, change the conclusions.

Agree. Conclusion reviewed. 

Reviewer 4 Report

The author has explored the effect of CF on AFC performance. Further research through relevant test methods has proved that it has certain research value for the fire resistance of materials, but there are still some deficiencies in the article, I leave some comments that I believe can help authors to improve the manuscript.

1、 Line 44, please check whether the logic of the sentence “Besides, all of the mentioned positive properties CF also has some drawbacks. Without a high price when compared with other fibers another important aspect is CF adhesion with cement matrix” is correct.

2、 Line 47, after reading the sentence “Seeking to increase …”, please determine whether the content behind the sentence is consistent with the sentence. Consider the following content is that how to enhance adhesion between CF and cement.

3、 Line 48, the word “cementing materials” should be consistent with the full test. In addition, please add some references after the sentence “Seeking to increase the adhesion in contact zone between CF and cementing materials, several methods are used: mechanical, chemical (modifying the fiber) and usage of polymeric additive.”.

4、 Line 49, it is an interesting sentence.

5、 Line 94, 102, 225. Please add labels a, b and c in the figure.

6、 The basic properties of cement materials should be displayed.

7、 Line 127, pressure: 10.13 or -10.13, temperature: 179oC or -179oC? To avoid ambiguity, please standardize the expression.

8、 Two different types of AFC were used, but the impact of the different types of fibers on the test is not mentioned. Please explain why these two types of fiber are used in this work.

9、 Please specify the order of adding materials.

10、 Line 192, the change rule can’t be clearly shown. In addition, as stated in line 190, is this change rule also caused by error?

11、 In Figure 5, the author describes the effect of CF on the ultrasonic pulse velocity, but dose not explain the reason for this. In addition, the trend is different after being affected by the fire effect. Please explain the reason for this phenomenon.

12、 Line 225 and 231, “Celulose fiber”, please check the spelling of the word.

13、 The change rule of compressive strength and bending strength is very interesting, please explain the reason for this phenomenon.

14、 Please compare at least 3 articles to determine the writing style and rewrite Part A.

15、 Line 297, “Ultrasonic pulse velocity measurements showed that up to high temperature effect increasing amount of CF in the samples, ultrasonic pulse velocity increases.”. Please determine whether the discussion is consistent with the change rule in Figure 5.

16、 After fire effect, the performance of the sample after adding CF decreases in some degrees. Through reading this article, I may think that the fire resistance of the sample without adding CF may be better. Please explain the reason and effect after adding CF in materials in detail.

Author Response

Dear Reviewer,
Thank You for comments and suggestions. We are sending for You revised manuscript and our answer for Your comments.  

  1. Agree. Corrected.
  2. Agree. Corrected.
  3. Agree. Corrected.
  4. Agree. Corrected.
  5. Agree. Added letters to figures.
  6. Agree. Basic properties added.
  7. Agree. Corrected.
  8. Revised. In manuscript authors mentioned that work with similar to industrial composition. We use similar proportions of different MFC. We started research with changing MCF to CF. For first step will be difficult use more complex changing to find dependence.
  9. Agree. Added the order of adding materials.
  10. Agree. Corrected.
  11. Agree. Explained in discussion. 
  12. Agree. Corrected.
  13. Agree. Explained in Discussion. 
  14. Agree. Part A was rewritten.
  15. Revised. Yes before fire UPV increase with increase CF adding, after fire exposure UPV decrease with increase CF adding. Revised sentence. 
  16. Agree. Added information in Discussion

Round 2

Reviewer 1 Report

The authors have addressed all my concerns. I have no further comments.

Reviewer 3 Report

The authors responded to all previous review comments.